# Development of a Low-Cost UV-Vis Spectrophotometer and Its Application for the Detection of Mercuric Ions Assisted by Chemosensors

**DOI:** 10.3390/s20030906

**Published:** 2020-02-08

**Authors:** David González-Morales, Asmilly Valencia, Astrid Díaz-Nuñez, Marcial Fuentes-Estrada, Oswaldo López-Santos, Olimpo García-Beltrán

**Affiliations:** 1Facultad de Ingeniería, Universidad de Ibagué, Carrera 22 Calle 67, Ibagué 730002, Colombia; david.gonzalez@unibague.edu.co; 2Facultad de Ingeniería Forestal, Universidad del Tolima, Altos de Santa Helena, Ibagué 730001, Colombia; ayvalencial@ut.edu.co; 3Universidad Nacional de Colombia, Sede Medellín, Escuela de Química, Carrera 65, No. 59A-110, Medellín 050034, Colombia; asdiazn@unal.edu.co; 4Facultad de Ciencias Naturales y Matemáticas, Universidad de Ibagué, Carrera 22 Calle 67, Ibagué 730002, Colombia; marcial.fuentes@unibague.edu.co

**Keywords:** water contaminants, low-cost spectrophotometer, Hg^2+^ ions, chemosensor, coumarin

## Abstract

Detection of an environmental contaminant requires the use of expensive measurement equipment, which limits the realization of in situ tests because of their high cost, their limited portability, or the extended time duration of the tests. This paper presents in detail the development of a portable low-cost spectrophotometer which, by using a specialized chemosensor, allows detection of mercuric ions (Hg^2+^), providing effective and accurate results. Design specifications for all the stages assembling the spectrophotometer and the elements selected to build them are presented along with the process to synthesize the chemosensor and the tests developed to validate its performance in comparison with a high-precision commercial laboratory spectrophotometer.

## 1. Introduction

A spectrophotometer is a measuring device for quantitative analysis generally used to characterize chemical substances by determining the amount of light that is partially absorbed by the analyte present in solution [1]. They can be classified according to the spectral region of work, such as ultraviolet spectrophotometer (UV), from 190 nm to 380 nm; visible spectrophotometer (Vis), from 380 nm to 750 nm; and near infrared spectrophotometer (NIR), from 800 nm to 2500 nm [2,3,4]. According to their use, they are classified in stationary devices for analysis in laboratories and portable devices for determination of substances in fieldwork [1].

There are spectrophotometers that work in non-visible ranges of the electromagnetic spectrum, such as those presented in References [5,6] (UV and NIR, respectively), which use high-cost elements like diode arrays and charged couple devices (CCD) as sensors in the detecting stage. For the visible range, there are several devices that include a Light Emitting Diode (LED) as radiation source [7], mainly because they have low costs and allow easy implementation. It is worth mentioning the developments presented in References [8,9] which made analyses for three fixed wavelengths and that reported in References [10,11,12] for seven fixed wavelengths which can be selected by the user with the help of mechanical components. Also, a combination of a white LED and interference filters can be used for obtaining different wavelengths as in Reference [13]. Additionally, some devices are used for chemistry education and allow the user to make a spectral sweep between 400 nm and 700 nm including a rotating diffraction grating [14]. In terms of portable equipment, those implemented in References [15,16,17,18,19,20,21] can also make spectral sweeps by means of the integration of small motors, light-scattering elements, low-cost light sources, and simple light detectors. Furthermore, there are commercial and robust devices for use in laboratory such as the Cary 60 UV-Vis (Agilent Technologies, CA, USA) [22], which allow the user to make many types of tests, obtaining high-quality analyses, selectivity, and accuracy at a cost of around 30,000 USD. Portable equipment, such as the DR 1900 (HACH, CO, USA) [23], provides versatility and other advantages in fieldwork at costs between 2000 USD and 8000 USD.

Over the last few decades, the impact of heavy metal pollution has been increasing [24]. Mercury is the heavy metal with the greatest impact on the pollution of ecosystems worldwide [25], contributing to bioaccumulation in different organisms and being present in the food chain [26]. Its oxidation state Hg^2+^ has a high affinity for sulfhydryl groups present in biomolecules such as proteins and endogenous thiols [27,28], which makes it possible to associate the accumulation of this metal to multiple physiological disorders, the best known of which are damages to the kidney, central nervous, immune, and endocrine systems [29,30,31,32,33,34]. There are a variety of analytical methods for determining mercury; however, the vast majority are complex and require a great deal of resources for implementation, maintenance, and sample analysis. Each of these methods has its own advantages and limitations. Among the best-known methods for determining mercury are hyper Rayleigh scattering (HRS), atomic absorption/emission spectrometry, inductively coupled plasma mass spectrometry (ICP-MS), colorimetric detection, cold vapor atomic fluorescence spectrometry, high-performance liquid chromatography, ion chromatography, and electrochemical sensing [35,36,37,38,39,40,41,42,43,44]. During the last decades, there has been increasing interest in developing chemical sensors that can interact with the analyte through desulphurization, chelation, mercury-mediated hydrolysis reactions, and inclusion in macromolecules assisted by chemosensors [24,27,35,45]. However, this tool complemented with low-cost UV-Vis spectrophotometry provides potential alternatives for countries which cannot have access to expensive instrumentation and sophisticated operating processes.

This paper presents the development of a portable low-cost spectrophotometer, which by using a chemosensor coumarin-derivate allows detection of mercuric ions (Hg^2+^) in an aqueous medium. The proposed device has important characteristics such as accuracy, low cost, and portability. The rest of the paper is organized as follows: Section 2 presents the components required to build the spectrophotometer, the required embedded digital control algorithms, the methods related with the synthesis of the chemosensors, and the process necessary to perform each test. After that, Section 3 presents results obtained using the proposed device in comparison with a commercial device and provides a discussion about how the effectiveness and accuracy of the built device enables its potential real application. Finally, conclusions are presented in Section 4.

## 2. Materials and Methods

### 2.1. Reagents

All analytes were purchased from Sigma-Aldrich and Merck and were used as received. The metal salts used were HgCl_2,_ CdCl_2_, PbCl_2_, CaCl_2_, MgCl_2_, ZnCl_2_, FeNH_4_(SO_4_)_2_·12H_2_O, Fe-(NH_4_)_2_(SO4)_2_·6H_2_O, CuCl_2_, MnCl_2_, and CoCl_2_. All solutions employed in this work were prepared in EtOH-H_2_O (20/80). ^1^H and ^13^C NMR spectra were recorded on an Avance II 400 MHz multidimensional spectrometer (Bruker Corporation, Billerica, MA, USA), using the solvent or the Tetramethylsilane (TMS) signal as an internal standard. High-resolution mass spectra were recorded on a Pegasus GC×GC (Leco Corporation, St. Joseph, MI, USA), GC-MS system with time of flight analyser, with 40,000 resolution. Absorption spectra were obtained by means of a Cary 60 spectrophotometer (Agilent Technologies, Santa Clara, CA, USA).

### 2.2. Chemosensor Development

The compound was prepared using a conventional three-step synthesis from commercial precursors until 7-(diethylamine)-2*H*-chromen-2-one (**2**) was obtained, followed by thionation with Lawesson´s reagent (Merck-Darmstadt, Germany.) as reported in the literature to obtain the compound 7-(diethylamino)-2*H*-chromene-2-thione (**3**) [46].

Synthesis of 7-(diethylamine)-2*H*-chromen-2-one (**2**) (9.8692 g, 87.84 %). ^1^H NMR (DMSO-*d*_6_) δ 7.50 (d, *J* = 8.0 Hz, 1H), 7.21 (d, *J* = 8.0 Hz, 1H), 6.53 (dd, *J* = 12.0 Hz, *J* = 2.0 Hz, 1H), 6.44 (d, *J* = 2.0 Hz, 1H); 5.99 (d, *J* = 8.0 Hz, 1H), 3.42 (q, *J* = 8.0 Hz, 4H), 1.17 (t, *J* = 8.0 Hz, 6H). ^13^C NMR (DMSO-*d*_6_): 162.7, 157.1, 151.1, 144.1, 129.2, 109.5, 109.07, 108.5, 97.9, 45.2, 12.8.

Synthesis of 7-(diethylamino)-2*H*-chromene-2-thione (**3**) (1.455 g, 62.47%). ^1^H NMR (DMSO-*d*_6_) δ 7.31–7.29 (m, 2H), 6.96 (d, *J* = 12, 1H), 6.66–6.64 (m, 2H). ^13^C NMR (DMSO-*d*_6_): 197.8, 159.6, 151.3 136.1, 128.9, 123.3, 110.8, 110.0, 97.3, 45.0, 12.4. High-resolution electrospray ionisation mass spectrometry (HRESIMS) was consistent with the molecular formula of C_9_H_14_OS (required: *m/z* obsd 233.0868; calcd for 233.0869).

### 2.3. Determining the Quantum Yield of Emission

Fluorescence quantum yield of **3** was measured using a solution of Rhodamine 101 in ethanol as standard (Φs = 1). All values were corrected considering the solvent refraction index. Quantum yield was calculated using Equation (1), where the subscripts x and s denote sample and standard, respectively; Ф is the quantum yield; η is the refractive index; and Grad is the slope from the plot of integrated fluorescence intensity vs. absorbance [35].
(1)$$\Phi_{x}=\Phi_{s}\left(\frac{Grad_{x}}{Grad_{s}}\right)\left(\frac{\eta_{x}^{2}}{\eta_{s}^{2}}\right)$$

### 2.4. Detection Limit

The detection of limit (LOD) was calculated based on absorbance and fluorescence titrations. To determine the signal to noise ratio, the absorbance (at 388 nm) of 3 with Hg^2+^ was measured three times and the standard deviation of calibration curve was determined. The detection limit was calculated with the equation LOD = 3 σ_b_/m, where σ_b_ is the standard deviation of calibration curve and m is the slope of the plot of absorbance or fluorescence intensity versus analyte concentration [35].

### 2.5. Electronic Components

The proposed low-cost spectrophotometer works in the visible part of the electromagnetic spectrum, from 400 nm to 700 nm (see Figure 1). It is composed by a low-power light source, a monochromator for selecting specific wavelengths, and two light-isolated sample chambers. Each sampling chamber has an independent detector which allows the spectrophotometer to make measurements of the reference and the studied substance in a single experiment.

#### 2.5.1. Light Source

Light Emission Diodes (LEDs) are widely used as light sources in analytical instrumentation applications [47]. For the proposed device, it is important to use a light source that emits radiation in the visible region of the electromagnetic spectrum. A 1-W low-cost LED with cold white color with a 5° collimator lens was selected, allowing the radiation pattern to be concentrated and reflected in a small portion of the diffraction grating, reducing any light power while the test is made. The relative intensity of a white LED shows that it has a good response in the visible region. Also, by using the additional hardware presented in Figure 1, the LEDs provide a practical way to have an economic alternative for the radiation source with low energy consumption (Appendix A).

#### 2.5.2. Monochromator

The main task of the monochromator is to select and transmit one specific wavelength of the visible spectrum to the sample. This goal is achieved in the proposed device using a National Electrical Manufacturers Association (NEMA) 17 hybrid stepper motor in micro-stepping configuration together with a low-cost diffraction grating and a small slit that is used for selecting wavelengths. The angle displacement of the diffraction grating is performed in steps of 1.8°, resulting in 200 steps for a full 360-degree turn. The Allegro a4988 controller is used in micro-stepping configuration to obtain a good resolution for sweeping all the wavelengths of the visible spectrum enlarging the number of steps. As the driver makes 1/16 of a step, the resolution results in 0.1125° per step. For details of the configuration of the driver, see Appendix A.

The diffraction grating splits and diffracts the radiation emitted by the white LED into individual wavelengths [48]. Unlike other dispersive elements such as mirrors and prisms, diffraction gratings are elements with good performance and lower costs. These are composed of a large number of small parallel slits that have equal spacing between them along the grating and which are capable of diffracting the incident light that falls on them. Depending on the overtone and the incident angle of the light beam of the source projected over the grating, an individual color can be observed. In the developed system, a diffraction grating obtained from a Digital Versatile Disc (DVD) without information is used. This is a low-cost solution, which contains grooves with a total of 1350 lines/mm and a separation between them of 780 nm, properties that provide good dispersion and resolution (see Table 1).

The diffraction grating is located in such a way that the range of displacement angles reflects in the samples the components corresponding to the third overtone produced by the reflection effect (Figure 2) with the aim of providing a greater spectral resolution in the visible range. Using the micro-stepping configuration available in the a4988 driver, in which one step is reduced to an advance of 0.1125°, the 300-nm spectrum is converted into 198 steps, resulting in a selectivity for the monochromator of 1.52 nm per step. To find the wavelength of a specific color, the following expression is used (Fraunhofer equation):(2)nλ=d sin θ,
where λ is the wavelength of a specific color; *n* = 1 (grating order); d = 740 (distance between slits in a DVD); and θ = diffraction angle.

Table 2 shows the values of the incident angle in the diffraction grating for different colors.

#### 2.5.3. Detector

The main function of a detector is to convert a light signal into an electric signal [49]. The TEMT6000 ambient light sensor (Vishay Intertechnologies, PA, USA) is used for the detection stage. This is a surface mount device that is composed of an Negative Positive Negative (NPN) phototransistor mounted on a printed circuit board (PCB) of 1 cm × 1 cm. Its relative spectral response is similar to the human eye, having good sensitivity in the visible range of the electromagnetic spectrum (Figure 3). The analog voltage output is proportional to the received amount of light. This voltage is conditioned before being sent to the microcontroller, using a non-inverting amplifier of gain 5. The detectors are located in such a way that they can receive the resulting monochromatic radiation after having passed through the sample and reference substances stored in the quartz cuvettes. Each position covered by the monochromator corresponds to a level of voltage in the detector, proportional to the light transmitted through the reference and sample studied substances.

Based on Beer Lambert’s law, the concentration value of a chemical substance is directly proportional to the amount of light it absorbs when exposed to monochromatic radiation. Once the voltage measurements corresponding to the light captured in the detector for the reference and the substance being studied are obtained, the following expression is used to find the absorbance value:(3)$$A=\log_{10}\frac{P}{\mathrm{Po}}\;=\;-\log_{10}\left(T\right)$$
where A is absorbance; P is magnitude of transmitted light through the sample; Po is magnitude of transmitted light through the reference; and P/Po = T, which is transmittance.

Using voltage levels, we have
(4)$$A=\log_{10}\frac{P}{P0}\;=\;-\log_{10}\left(\frac{\mathrm{Vsample}}{V\;\mathrm{solvent}}\right)$$

Table 3 presents the most absorbed color in every produced wavelength.

#### 2.5.4. Microcontroller and Control Algorithms

An Atmega 138p microcontroller (Atmel Corp., San Jose, CA, USA) with an Arduino interface was used to implement the control algorithms required by the spectrophotometer operation. First, an initialization routine is performed, which consists in finding an initial position for the monochromator and using a physical reference with a limit switch that guarantees that all tests start at the same wavelength. Then, using micro-stepping, the stepper motor sweeps all the wavelengths of the visible spectrum. In each step, the voltage value produced by the detector is obtained, measuring the resulting monochromatic light that has passed through the sample. This voltage is read by the analog-to-digital converter of the microcontroller, where it is filtered using a simple average digital filter. After that, the absorbance value is computed and stored in a table. The flowchart of the developed algorithm is presented in Figure 4.

#### 2.5.5. Detailed Hardware Setup

The components are assembled in a black box made with polylactic acid (PLA) polymer by means of 3D printing with a size of 15 cm × 22 cm × 8 cm and a wall and top cover thickness of 4 mm, ideal to reduce the incidence of external light interferences. In order to obtain good light dispersion, the diffraction grating is located 18 cm from the sample chambers. In addition, it has the ability to rotate its position to make incident angles between 25° and 70°, which guarantees the generation of monochromatic light between 400 nm and 700 nm. The detectors are in a room with light insulation that reduces the interference in the experiments and guarantees zero voltage when the radiation source is off. Figure 5 shows the physical distribution of the components.

#### 2.5.6. Tests Applied to the Developed System

To test the operation of the developed equipment, an initial test is performed to measure the concentration of the sensor with coumarin, as is done in Reference [50]. A solution with ethanol is used as reference, and 4 solutions with coumarin in 4 known concentrations are tested: 5 μM, 10 μM, 15 μM, and 20 μM. The second test consists in measuring mercury concentration. Different solutions are prepared with known concentrations of 1 μM, 2 μM, 3 μM, and 4 μM, which are added to the coumarin with the sensor. Figure 6 shows the color change effect in coumarin when detecting mercury. The tests run in parallel, making comparisons between the results obtained with the developed prototype and a commercial Cary 60 spectrometer (Agilent Technologies, Santa Clara, CA, USA).

## 3. Results and Discussion

When building the low-cost UV-Vis spectrophotometer, consideration was given to the development of a chemosensor to evaluate the optical capabilities of the proposed equipment, which led to a search in the literature to find alternatives for its synthesis. According to that, the developed chemosensor must have the characteristic to react and determine the metal in aqueous medium by colorimetric and spectrophotometric methods.

Compound **3** was prepared in three synthetic steps (Appendix A). 4-(diethylamino)-2-hydroxybenzaldehyde (**1**) was condensed with diethyl malonate in a Knoevenagel condensation and cyclized and decarboxylated in one step to afford 7-(diethylamino)-2*H*-chromen-2-one (**2**). Then, the compound was modified with Lawesson’s reagent (thionation agent) to obtain the compound 7-(diethylamino)-2-chromene-2-thione (**3**) (Figure 6) [46]. The final product was characterized by using ^1^H and ^13^C Nuclear Magnetic Resonance (NMR) spectroscopy and High Resolution Mass Spectrometry (HRMS) (Appendix A).

The main photophysical constants of compound **3** were determined, showing an excitation band (λ_ex_) at 483 nm (λ_ex_) and an emission band (λ_ex_) at 510 nm (Appendix A); a molar extinction coefficient, which is 15,754 M^−1^ cm^−1^ (Appendix A); a quantum yield of 0.013; and a Stokes shift of 27 nm.

Compound **3** was tested with different metals (Hg^2+^, Cd^2+^, Pb^2+^, Ca^2+^, Mg^2+^_,_ Zn^2+^, Fe^2+^, Fe^3+^, Cu^2+^, Mn^2+^, and Co^2+^). When testing metals, it is observed that only mercuric ions react with chemosensor 3 very quickly and selectively, showing a colorimetric (Figure 7a), fluorimetric (Figure 7b), and spectrophotometric UV-Vis (Figure 8). In reviewing the phenomenon that occurred in the latter result being notorious, by adding mercuric ions in the solution of compound **3**, there is a decrease in the band of 483 nm and a bathochromic displacement to 513 nm; then, over time, it is observed that the 513-nm band decreases and that there is a hypochromatic shift with an increase of one band in 388 nm UV-Vis [46] (Appendix A) and an isosbestic point at 432. Using different concentrations of mercuric ions (0.25 to 2.5 µM), the detection limit was calculated as 1.1 × 10^−9^ M and a limit of quantification was calculated as 3.7 × 10^−9^ M. In addition, we emphasize that, when testing with the addition of mercuric and other ions simultaneously, no interference was observed.

To confirm and elucidate what was happening in the reaction, we conducted an experiment for the ^13^C NMR (Appendix A). When compound **3** was performed in the absence and presence of mercuric ions, a desulphurization reaction was evident [35]. By adding Hg^2+^, the OC = S carbon atom C2 of the thiocoumarin (δ = 194.7) was clearly displaced to 166.0 ppm, a typical C2 carbonyl (OC = O) signal of a coumarin (Appendix A).

By validating the response of the Cary 60 spectrophotometer (Agilent Technologies, Santa Clara, CA, USA) and the built low-cost spectrophotometer, using both equipment, the length of absorption was determined reporting a maximum of at 483 nm. In Figure 9, it can be clearly seen that the absorption bands are very similar between both instruments. This gives a clear idea of the approach in the correct operation of the built device. Although the response presented by the low-cost spectrophotometer has some noise compared to the Cary 60 spectrophotometer (Agilent Technologies, Santa Clara, CA, USA) and it should be further reduced, this will imply that additional electronic circuitry or additional computational requirements have a direct impact on the cost of the device or the duration of the test, respectively.

In contrast, Table 4 shows the comparison between the measurements made using both devices. It is important to note the absorbance measurements with error percentages lower than 10%, which are similar in both instruments. The measurement with the error percentage higher than 10% is obtained in the lowest coumarin concentration, which is seen as an opportunity to improve the sensitivity of the detectors so that they have better performances at these specific conditions.

With the obtained results, the calibration curve of the instrument for the case of coumarin concentration is computed. This is presented in Figure 10.

Just like the previous test, both the prototype and the commercial device show the same absorption band characteristics as compound **3**; by adding different concentrations of mercuric ions, the increasing behavior of a new absorption band is evident. This displacement occurs by a desulphurization reaction [35] when Hg^2+^ reacts with the sulfur of thiocarbonyl of compound **3**, additionally and perceptible changing to the human eye given colorimetric changes of yellow to pink. Likewise, it can be seen in Figure 11 how the signal forms are very similar between the reported in both instruments. There are some variations that are identified in the lower part of the signal given by the prototype, which can be corrected by implementing additional filters.

Table 5 shows the comparison between the results obtained from both compared devices. Percentage errors in the absorbance measurements are less than 5%, which shows the accuracy in the detection of the concentration of mercury ions.

With the obtained results, the calibration curve of the instrument for the case of mercury concentration is computed. This is presented in Figure 12.

In addition to its good performance, the total cost of the device deserves special attention since it is lower than 86 USD. Cost discriminated by components is presented in Table 6.

## 4. Conclusions

A portable low-cost spectrophotometer was developed and described in detail in this paper. The use of the proposed device has been focused on but is not limited to detection of mercuric ions Hg^2+^ in an aqueous medium. A particular feature to highlight of the proposed device and its test method is the use of a chemosensor which allowed the determination of mercuric ions in micromolar concentrations by means of a desulphurization reaction which was characterized by spectrophotometric methods and through ^13^C NMR spectroscopy. When evaluating the developed device, only slight differences were obtained between the results of the proposed device and the equivalent equipment used in a traditional laboratory. Future efforts will concentrate on extending the possible application of the device to analysis of quality in fuels, determination of plasma glucose, color measurement, and even forensic applications. This work demonstrates how interdisciplinary interaction between electronics engineering and basic sciences can provide solutions facilitating a wide research and academic community having access to specialized test like this.

## Figures and Tables

**Figure 1 sensors-20-00906-f001:**
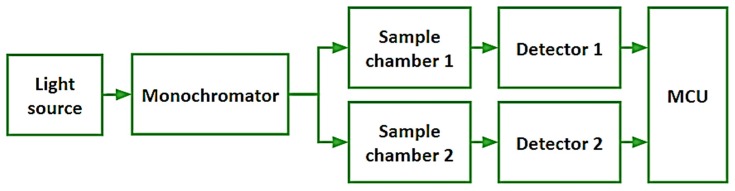
System architecture of the developed spectrophotometer.

**Figure 2 sensors-20-00906-f002:**
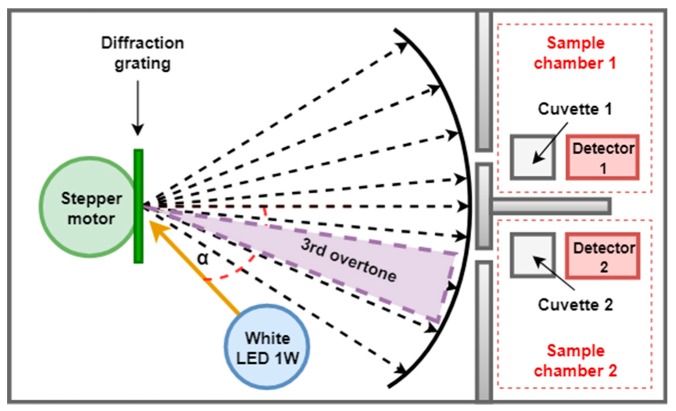
Selected overtone produced by the diffraction grating.

**Figure 3 sensors-20-00906-f003:**
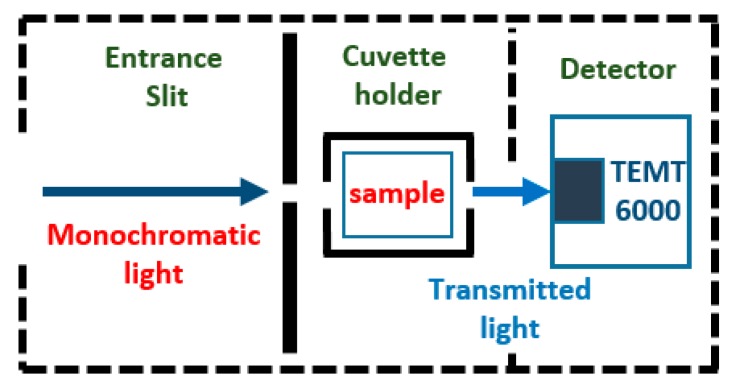
Details of the detector stage.

**Figure 4 sensors-20-00906-f004:**
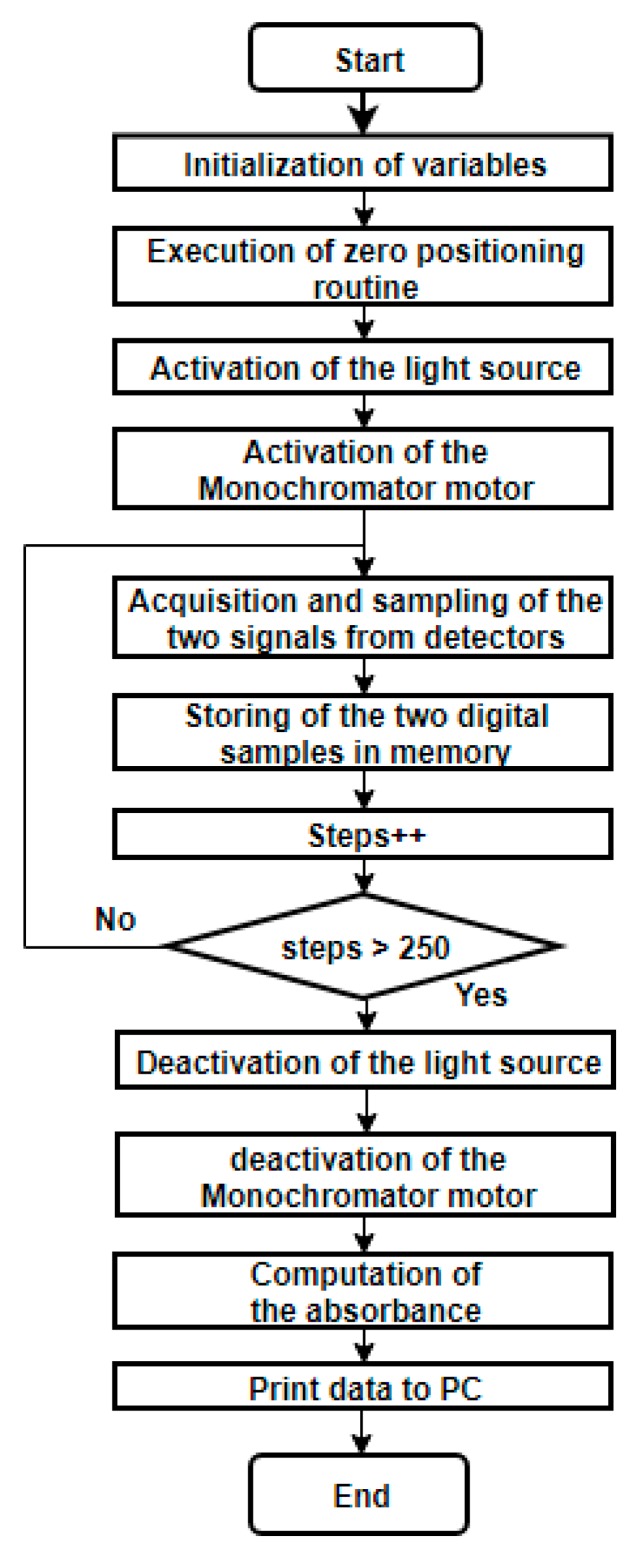
Flowchart of the microcontroller algorithm.

**Figure 5 sensors-20-00906-f005:**
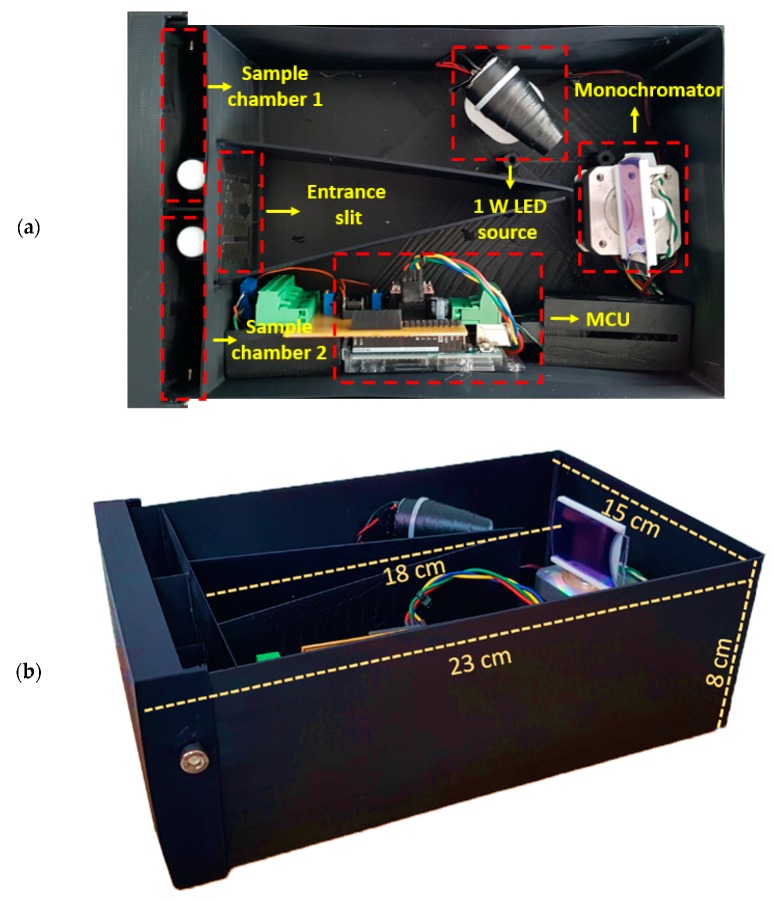
(**a**) Geometric distribution of the spectrophotometer hardware; (**b**) dimensions of the developed system.

**Figure 6 sensors-20-00906-f006:**
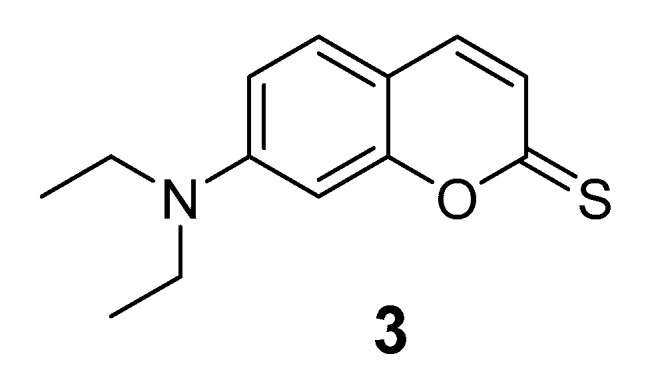
Chemical structure of 7-(diethylamino)-2*H*-chromene-2-thione.

**Figure 7 sensors-20-00906-f007:**
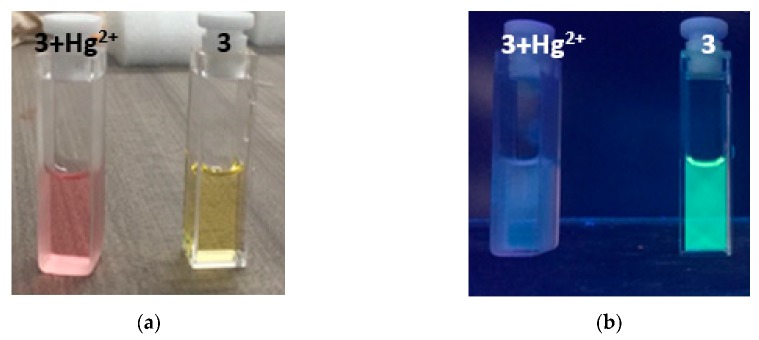
(**a**) Colorimetric change of compound **3** in the presence of mercuric ions; (**b**) fluorimetric change of compound **3** in the presence of mercuric ions.

**Figure 8 sensors-20-00906-f008:**
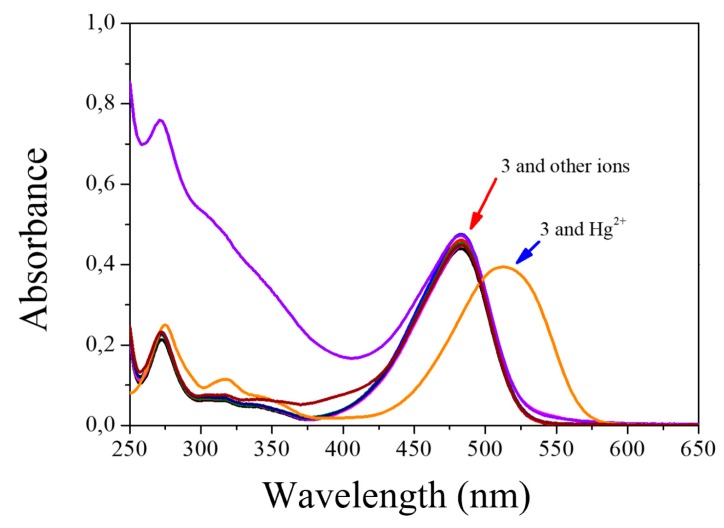
Changes in the absorbance spectra of **3** (5 × 10^−6^ M) upon addition of ions (1 × 10^−4^ M) in H_2_O:EtOH 80:20.

**Figure 9 sensors-20-00906-f009:**
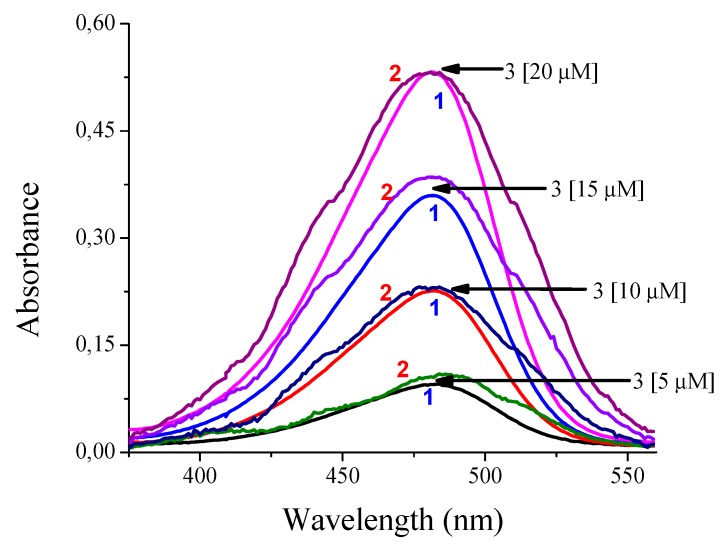
Comparison of the absorbance responses of compound **3** using (**1**) the spectrophotometer Cary 60 and (**2**) the low-cost spectrophotometer in H_2_O:EtOH 80:20.

**Figure 10 sensors-20-00906-f010:**
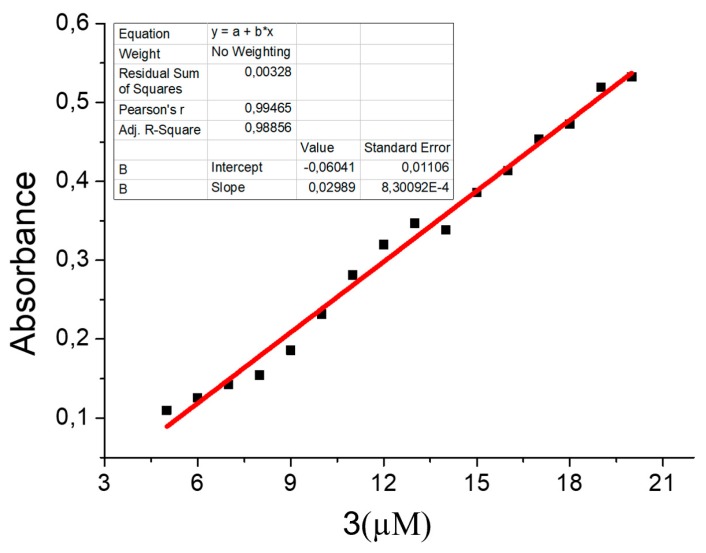
Calibration curve for coumarin concentration.

**Figure 11 sensors-20-00906-f011:**
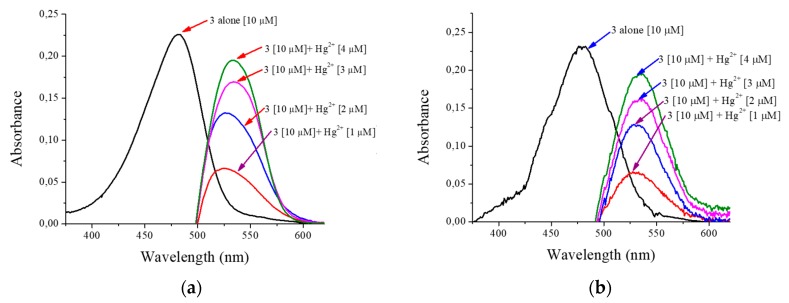
Absorbance the compound **3** and response in front mercuric ions: (**a**) spectrophotometer Cary^®^ 60; (**b**) the low-cost spectrophotometer in H_2_O:EtOH 80:20.

**Figure 12 sensors-20-00906-f012:**
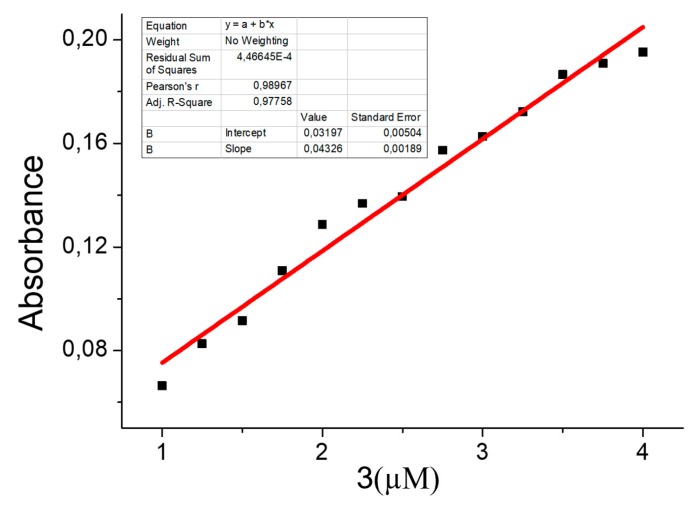
Calibration curve for Hg^2+^ at different concentrations.

**Table 1 sensors-20-00906-t001:** Grating separation for different diffraction gratings.

Grating	Lines (inch)	Lines (mm)	Grating Separation (nm)
CD	15,875	625	1600
DVD	34,300	1350	740
Lab grade diffraction grating	34,300	1000	1000

**Table 2 sensors-20-00906-t002:** Incident angle in the diffraction grating for obtaining different colors.

Color	Wavelength (nm)	Deflection Angle (°)
red	650	61
orange	600	54
yellow	575	51
green	550	48
blue-green	500	42.5
blue	450	37.5
violet	400	32.7

**Table 3 sensors-20-00906-t003:** Light absorbance for wavelengths in visible spectra.

Wavelength	Absorbed Color	Solution Color
380–435	Violet	Green-yellow
435–480	Blue	Yellow
480–490	Blue-green	Orange
490–500	Green-blue	Red
500–560	Green	Violet
560–580	Green-yellow	Violet
580–595	Yellow	Blue
595–650	Orange	Blue-green
650–780	Red	Green-blue

**Table 4 sensors-20-00906-t004:** Measurements of coumarin concentration.

Concentration	Absorbance Prototype	Absorbance Cary 60	Measurement Difference	Error Percentage (%)
5 µM	0.109	0.095	0.014	12.84
10 µM	0.232	0.226	0.006	2.65
15 µM	0.385	0.360	0.025	6.94
20 µM	0.532	0.532	0.000	0.0

**Table 5 sensors-20-00906-t005:** Measurements of mercury concentration.

Concentration	Absorbance Prototype	Absorbance Cary 60	Measurement Difference	Error Percentage (%)
1 µM	0.066	0.065	0.001	1.54
2 µM	0.128	0.132	0.004	3.03
3 µM	0.162	0.169	0.007	4.14
4 µM	0.195	0.194	0.001	0.52

**Table 6 sensors-20-00906-t006:** Total cost of the developed spectrophotometer.

Component	Cost (USD)
Light source	2
Stepper motor	14
Arduino UNO	21
Stepper driver	4
Light detectors	10
Black Box	20
Additional elements	15
Total	86

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
