# Peer review of "Development of a Low-Cost UV-Vis Spectrophotometer and Its Application for the Detection of Mercuric Ions Assisted by Chemosensors"

_sensors, 2020, doi:10.3390/s20030906_

Round 1
Reviewer 1 Report
Beltrán et. al reported work entitled on "New Low cost UV-Vis spectrophotometer for the detection of Mercuric ions assisted by chemosensors" is interesting however this work can be improved with few corrections as I have mentioned below.
1) There are few typographical errors in the manuscript for example in page no:4 line 89 to 98. Synthesis of ??. H-RMN ?? What is it means ? does it NMR notation? I strongly urge to authors to use standard notations for the characterisation of compounds.
2) Please incorporate the Mass spectra of compound in synthesis part.
3) Please incorporate the structure of respective coumarin based chromogenic materials with appropriate scientific notations.
4) Can authors explain why they have used comarine core for the detection of Hg2+ there are several chromogenic materials were commercially available that could detect the Hg2+ ions very effectively with zero background signals. If there is an significant reason or relationship with constructed new portable instrument ? author should elaborate it.
4) How it is different from single beam UV-vis spectrophotometer ? Only in light source or other factors like without reference chamber etc.
5) It is better to keep the lowest possible response (limit of detection and limit of quantification) using current instrument.
Author Response
Reviewer 1
Beltrán et. al reported work entitled on "New Low cost UV-Vis spectrophotometer for the detection of Mercuric ions assisted by chemosensors" is interesting however this work can be improved with few corrections as I have mentioned below.
There are few typographical errors in the manuscript for example in page no:4 line 89 to 98. Synthesis of ??. H-RMN ?? What is it means ? does it NMR notation? I strongly urge to authors to use standard notations for the characterisation of compounds.
Answer: We appreciate the reviewer's comments, as the changes have already been made to the document. The abbreviation is NMR and not as written.
Please incorporate the Mass spectra of compound in synthesis part.
Answer: We appreciate the reviewer's comments, the suggestions were incorporated into the manuscript and supporting data.
3) Please incorporate the structure of respective coumarin based chromogenic materials with appropriate scientific notations.
Answer: We appreciate the reviewer's comments, the suggestions were incorporated into the manuscript between the lines 222 to 223.
4) Can authors explain why they have used comarine core for the detection of Hg2+ there are several chromogenic materials were commercially available that could detect the Hg2+ ions very effectively with zero background signals. If there is an significant reason or relationship with constructed new portable instrument? author should elaborate it.
Answer: Our group has been working for about 8 years in the development of chemosensors for organic molecules, metals of biological and environmental interest, and we have used coumarin (2H-chromen-2-one) as the chemical core. For example, we have recently published the article:
A selective thioxothiazolidin-coumarin probe for Hg2+ based on its desulfurization reaction. Exploring its potential for live cell imaging. Spectrochimica Acta Part A: Molecular and Biomolecular Spectroscopy, 224, (2020) 117372.
Where we also report a chemosensor based on disulfurization reactions.
We also clarify that there is no direct relationship between the device and the chemosensor. This equipment can work independently with other materials that have an absorption interval between 400 and 700 nm.
4) How it is different from single beam UV-vis spectrophotometer? Only in light source or other factors like without reference chamber etc.
Answer: This is a single beam spectrophotometer. The main difference whit other developments is that this equipment has two chambers in which the two samples (reference and substance to be analyzed) are stored. These chambers are located inside the device in such a way that a single sweep of the monochromator allows to obtain the required measurements.
In order to clarify this feature, the sample chambers were identified in figure 1, 2 and 5. Mentions about sample chambers were clarified in pag. 3 line 102 and pag. 6 line 194.
5) It is better to keep the lowest possible response (limit of detection and limit of quantification) using current instrument.
Answer: We agree with the reviewer comment because the scope of the paper is limited to the application of the proposed device for the detection of mercuric ions in an aqueous medium. The corresponding clarifications were done in the manuscript in pag. 10 lines 279-280 and the conclusions section. Hovewer, the detection and quantification limit values for the experimental conditions for compound 3 used in this work have been incorporated.

Reviewer 2 Report
Manuscript ID sensors 680508
The manuscript entitled “Development of a Low-Cost UV-Vis Spectrophotometer and its Application for the Detection of Mercuric Ions Assisted by Chemosensors” is quite interesting as it provides a detailed description of the instrument manufactured and execution of the test results using the low-cost spectrometer. The first part of the manuscript that describes the making of low-cost instrument is an already published material (ref. 50 of the article) by the authors, González-Morales et al. Details on the sensing activity is a new addition.
Here are some comments to the authors to consider:
The devised instrument has comparative absorbance as of Cary 60, however, the instrumental noise is not improved from ref. 50. I would like the authors to comment on the signal: noise ratio of the low-cost instrument and compare with Cary 60. The coumarin based compound used in the present work is the sensor to the Hg2+ ion. I would like the authors to compare the selectivity of the output compared to the other ions used for screening in presence of Hg2+ ion. After the interaction of Hg2+ with the sensor, the authors commented on the formation of a new compound where desulfurization occurs and C=S is replaced by C=O? I would like the authors to confirm the formation of 7-(diethylamino)-2H-chromen-2-one with the observed absorption of the compound 3 after reaction with Hg2+. Determination of binding of the sensor with Hg2+ ion and measurement of the detection limit is worthwhile to improve the manuscript. Figure 5, the sample chamber is not indicated. Figure S7 in the manuscript does not match with the one described. Pg. 2, Line 44, rephrase. Look for grammatical errors and spellings in pgs. 2, 3, 6 (flowchart).

Author Response
Reviewer 2
The manuscript entitled “Development of a Low-Cost UV-Vis Spectrophotometer and its Application for the Detection of Mercuric Ions Assisted by Chemosensors” is quite interesting as it provides a detailed description of the instrument manufactured and execution of the test results using the low-cost spectrometer. The first part of the manuscript that describes the making of low-cost instrument is an already published material (ref. 50 of the article) by the authors, González-Morales et al. Details on the sensing activity is a new addition.
Here are some comments to the authors to consider:
The devised instrument has comparative absorbance as of Cary 60, however, the instrumental noise is not improved from ref. 50. I would like the authors to comment on the signal: noise ratio of the low-cost instrument and compare with Cary 60.
Answer: The comparison proposed by the reviewer is very interesting, however, it involves a complex procedure since estimation or measurement of noise ratio implies complex theoretical analysis or measurements. Please take into account that the purpose of this paper is limited to the detection of a specific metal for which the results presented clearly show that the SNR is acceptable.
Furthermore, as mentioned by the reviewer, the instrument provides a similar absorbance measurement as the Cary 60 which is a high precision instrument used in laboratory tests. Therefore, although it is possible to improve this performance indicator, this implies additional electronic circuitry or additional computational requirement which has a direct impact on the cost of the device or the duration of the test respectively.
To clarify this aspect, the following line was included in page 9 of the document:
“Although, the response presented by the low-cost spectrophotometer have some noise compared to Cary 60 spectrophotometer and it should be further reduced, this will imply additional electronic circuitry or additional computational requirement which has a direct impact on the cost of the device or the duration of the test respectively”.
The coumarin based compound used in the present work is the sensor to the Hg2+ I would like the authors to compare the selectivity of the output compared to the other ions used for screening in presence of Hg2+ ion.
Answer: Thanks to the reviewers for their suggestions, we have included the recommendations in the letter between lines 232 and 240.
After the interaction of Hg2+ with the sensor, the authors commented on the formation of a new compound where desulfurization occurs and C=S is replaced by C=O? I would like the authors to confirm the formation of 7-(diethylamino)-2H-chromen-2-one with the observed absorption of the compound 3 after reaction with Hg2+.
Answer: In response to the reviewer, we confirmed the presence of 7-(diethylamino)-2H-chromen-2-one by taking NMR spectra 1H and 13C. In the support material (Figure S9) we compared the 13C NMR of the compound 7-(diethylamino)-2-chromene-2-thione (3) without mercuric ions in the upper part of the graph and in the lower part the 13C NMR of the reaction of 7-(diethylamino)-2-chromene-2-thione (3) with mercuric ions. What was observed and interpreted corresponds to the spectrum of 13C NMR of 7-(diethylamino)-2H-chromen-2-one. In addition, it was also tested using UV-Vis spectrophotometry as seen in the following graph, included in the support material.
Determination of binding of the sensor with Hg2+ ion and measurement of the detection limit is worthwhile to improve the manuscript.
Answer: we appreciate the indications of the reviewers to improve the manuscript, the detection and quantification limit values for the experimental conditions used in this work have been incorporated.
Figure 5, the sample chamber is not indicated.
Answer: The figure was modified according to this recommendation.
Figure S7 in the manuscript does not match with the one described.
Answer: Thanks to the reviewers for their suggestions, the writing was modified according to this recommendation.
2, Line 44, rephrase.
Answer: This paragraph was modified according to this recommendation.
Look for grammatical errors and spellings in pgs. 2, 3, 6 (flowchart).
Answer: These pages were revised according to this recommendation.

Reviewer 3 Report
Authors did good work and can be published after minor revisions. The issues to be considered as follows.
1. Line 97, "should be corrected as Synthesis of 3".
2. Mass spectral data should be provided.
3. Figure 9 and 11, calibration curve should include more points.
4. What was the reason to choose EtOH:H2O and why in 80:20. Authors should explain it clearly
5. Is there any indication of resonance energy transfer ?
6. What is the mechanism the observed mercury sensing follows ? Static or dynamic ?
7. Fluorescence life time experiments should be provided.
Author Response
Reviewer 3
Authors did good work and can be published after minor revisions. The issues to be considered as follows.
Line 97, "should be corrected as Synthesis of 3".
Answer: We appreciate the reviewer's comments, the suggestions were incorporated into the manuscript
Mass spectral data should be provided.
Answer: We appreciate the reviewer's comments, the suggestions were incorporated into the manuscript and supporting data.
Figure 9 and 11, calibration curve should include more points.
Answer: We appreciate the reviewer's comments, the suggestions the figure 9 and 11 were incorporated into the manuscript
What was the reason to choose EtOH:H2O and why in 80:20. Authors should explain it clearly
Answer: First we want to make it clear to the reviewer that the mixture used is EtOH:H2O 20:80, allowed us excellent solubility without using other solvents such as DMSO or buffers and the results obtained allowed us to calculate interesting detection limits.
Is there any indication of resonance energy transfer?
Answer: This type of coumarinic compound has functional groups such as diethylamine, which is a strong electronic giver and electro-acceptor group like C-2 carbonyl. It is expected that by modifying with S its electro-attractive capacity will be increased, displaying strong intramolecular charge transfer (ICT).
What is the mechanism the observed mercury sensing follows ? Static or dynamic ?
Answer: According to what has been observed and taking into account the reaction speeds, the predominant mechanism is the dynamic.
Fluorescence life time experiments should be provided.
Answer: Infortunately we do not have the instrumentation to respond to this reviewer's request.

Round 2
Reviewer 2 Report
The authors have significantly modified the manuscript based on the comments. I will recommend publishing at its present form.